# Effect of music therapy on behavioral and physiological neonatal outcomes: A systematic review and dose-response meta-analysis

**Fatemeh Shahbazi**[1,2], **Marzieh Fattahi-Darghlou**[1], **Samad Moslehi**[3], **Minoo Dabiri-Golchin**[4], **Marjan Shahbazi**[5]*

1 Department of Epidemiology, School of Public Health, Hamadan University of Medical Sciences, Hamadan, Iran, 2 Modeling of Noncommunicable Diseases Research Center, Health Sciences & Technology Research Institute, Hamadan University of Medical Sciences, Hamadan, Iran, 3 Department of Biostatistics, School of Public Health, Hamadan University of Medical Sciences, Hamadan, Iran, 4 Department of Occupational Therapy, University of Manitoba, Winnipeg, MB, Canada, 5 Department of Occupational Therapy, School of Rehabilitation Sciences, Iran University of Medical Sciences, Tehran, Iran

* marjanshahbazi55@yahoo.com

## Abstract

### Background

Previous studies have documented the effectiveness of music therapy in improving adverse neonatal outcomes in premature infants. However, this review aims to address the question of how long listening to music can enhance these neonatal outcomes.

### Methods

To conduct this dose-response meta-analysis, we searched the PubMed, Scopus, Web of Science, and Cochrane Library databases. The inclusion criteria comprised randomized clinical trials that investigated the effects of music therapy on improving adverse neonatal outcomes. Preterm infants were defined as those born between 27 and 37 weeks of gestation, as fetuses are known to respond to auditory stimuli starting at the 27th week of pregnancy. The initial search was performed on January 28, 2024, and there were no restrictions on the time frame for the search. Ultimately, we employed a two-stage random effects model using the "drmeta" package in Stata software to perform this dose-response meta-analysis.

### Results

In total, 30 articles (1855 participants) were identified for inclusion in our meta-analysis. According to pooled analysis with each minute increase in music therapy, the means of respiratory rate, pain score, SBP, DBP, behavioral score, and body temperature decrease by 35.3 beats per minutes, 15.3 VAS, 30.7 mmHg, 8.9 mmHg, 2.7, and 0.27˚C. On the other hand, with each minute increase in listening to the music, the mean of O2 saturation, heart rate and sleep duration increase 1.7%, 89.2 beats per minutes and 5.081 minutes per day, respectively.

**Data Availability Statement:** All relevant data are within the paper and its Supporting Information files.

**Funding:** The author(s) received no specific funding for this work.

**Competing interests:** The authors have declared that no competing interests exist.

## Conclusion

Music therapy improves the neonatal outcomes of O2 saturation, heart rate, respiratory rate, sleep duration, body temperature and systolic and diastolic blood pressures. Therefore, the existence of a dose-response relationship can indicate a causal relationship between music therapy and the improvement of neonatal outcomes.

## Introduction

Music therapy is the clinical and evidence-based use of music interventions with organized melody, rhythm, harmony, and timbre to improve health outcomes [1, 2]. Music is often considered an auxiliary medical treatment due to its involvement in physiological, psychological, and social functions [3–5]. It can also influence the nervous system's cortical, subcortical, and vegetative areas, strengthen psycho-physiological defenses, and improve health [6]. The benefits of Music therapy for treating adverse neonatal outcomes have been investigated in previous research [7, 8]. Music therapy has shown the ability to improve and stabilize newborns' vital signs and physiological responses [4, 9]. This intervention can improve premature babies' sleep duration, body temperature, oxygen arterial saturation, and behavioral scores. It can also reduce pain and anxiety levels [10].

In recent years, clinicians and neonatologists have been using music therapy for the reasons of low cost, being non-invasiveness, and non-pharmacological for premature infants hospitalized in the NICU [10, 11]. Based on the findings of previous single studies, music therapy has played an effective role in improving various conditions of premature neonates such as heart rate, breathing rate, oxygen saturation, and the functioning of the parasympathetic nervous system [12]. Even in some studies, it was found that combining it with some care methods such as kangaroo care has relieved the pain of infants; however, the results are inconsistent [13, 14]. A meta-analysis in 2012 showed the effectiveness of music in infants admitted to the ICU, but this study had two serious limitations: first, it included only articles that were published in English, and second, the included studies were mostly observational not randomized clinical trials [15]. Recently, Yue et al published a systematic review and meta-analysis and concluded that music therapy can significantly effect on improving the heart and respiratory rates and stress level of premature infants [10]. The recent meta-analysis only examined the outcomes of heart rate, respiratory rate, oxygen saturation, behavioral state, and stress level, while the present meta-analysis investigate nine physiological and behavioral neonatal outcomes. On the other hand, after the latest meta-analysis, new randomized clinical trials have been published on music therapy and neonatal conditions; therefore, an updated systematic review and meta-analysis is needed. In addition, this synthesis focused on dose–response relationship between massage therapy and neonatal conditions for the first time. Consequently, this systematic review examines the dose-response relationship between music therapy and neonatal outcomes in preterm infants.

## Method

### Eligibility criteria (PICOS)

**Population.** The study population comprised preterm infants who were born between 27 and 37 weeks of gestation (fetuses respond to auditory stimuli from the 27th week of pregnancy). Intervention: The intervention group included neonates who were hospitalized in the neonatal intensive care unit (NICU) and received music therapy.

**Control.**　The control group consisted of neonates who received usual care in NICU.

**Outcome.**　The primary outcome was neonatal status. Neonatal status included heart rate (beats per minute), systolic and diastolic blood pressures (mmHg), respiratory rate (breaths per minute), pain score (visual analog scale (VAS)), behavioral score, sleep duration (hours per day), oxygen arterial saturation (%), and body temperature (˚C). Also, music therapy is an auxiliary and palliative treatment, and according to the literature review, there were no secondary outcomes and side effects to reporting.

**Studies.**　Randomized controlled trials were included in the analysis regardless of language and their publication status. These studies specifically investigated the effectiveness of music therapy on neonatal status. To be eligible for inclusion, studies were required to present the mean difference (MD) along with the corresponding 95% confidence interval (CI), or provide sufficient information to calculate the effect sizes manually.

**Confounding control.**　We included randomized controlled trials. Randomization in clinical trials is a method of adjusting confounder variables in experimental studies. In other words, randomization makes the studied groups to be balanced in terms of known and unknown confounders. For this reason, we only examined the experimental studies that investigated the effects of music therapy with a randomized design.

**Handling missing data.**　The effect size examined in this study was the mean difference. In several of the included articles, the mean difference in behavioral and physiological indicators between the intervention and control groups was clearly reported. In other studies, the mean behavioral and physiological outcomes for infants receiving music compared to the control group were reported at the end of trial; in these cases, we manually calculated the mean difference. In general, the studies included in the final analysis did not have missing data that would hinder the calculation of the mean difference. Also, if the full text of an article is not available, we would submit a request for access on the Iranian site Iran Paper, allowing the article to be provided to the author within 24 hours upon payment.

**Information source and search strategy.**　We searched four electronic databases for eligible studies: PubMed, Scopus, Web of Science, and Cochran Library. The search was conducted on Juan 28, 2024, and its start date was unlimited. We searched varying combinations of keywords and Mesh terms (((((((((((premature infant) OR (infant, premature)) OR (prematurity, neonatal)) OR (LBW)) OR (low birth weight)) OR (premature infant*)) OR (preterm infant*)) OR (prematurity)) AND ((((((music therapy) OR (auditory stimulation)) OR (acoustic stimulation)) OR (music)) OR (song*)) OR (singing*)) OR (therapy, music))) AND ((((randomized controlled trial)) OR (RCT)) OR (placebo)) OR (random)). We also manually screened the reference lists of all relevant articles to find additional articles that might have been missed in the systematic search. Duplicate reports of the same study were deleted.

**Selection and data collection process.**　We used EndNote software (version X8) for Study screening and selection. Two investigators (MF and MSH) independently screened all non-duplicate titles and abstracts, evaluated the full texts for eligibility, and performed data extraction using a predesigned data form in Stata software. Any discrepancies between authors were resolved by discussion or consensus with an expert (FSH).

**Data items.**　The data extracted from all included studies were author's first name, publication year, outcome, country, number of participants, mean difference, standard deviation, type of trial (parallel RCTs and crossover RCTs), type of music (live or recorded music), and duration of music playback during the trial (per minutes in day). Details of the extracted data are presented by study in S1 Dataset.

**Determining the dose of music.**　To determine the dose of music for each study, we first decided how many days the preterm infants received music therapy in each trial. In the next step, we determined how many minutes they listened to music each day. In the last stage, the

final dose was determined by multiplying the number of intervention days by the number of minutes of listening to music.

**Quality assessment of studies.** We used the Cochrane criteria for RCT studies to assess the methodological quality of the included studies. We evaluated the quality of all relevant studies based on random sequence generation (selection bias due to inadequate generation of a randomized sequence), allocation concealment (selection bias due to insufficient conceal-ment of allocations before assignment), blinding of participants and personnel (performance bias due to knowledge of the allocated interventions by participants and personnel during the study), blinding of outcome assessment (detection bias due to knowledge of the allocated interventions by outcome assessor), incomplete outcome data (attrition bias due to amount, nature or handling of incomplete outcome data), and selective reporting (reporting bias due to selective outcome reporting). Then, the included RCTs were classified as low risk of bias if all mentioned criteria were met, unclear risk bias if one item was not met, and high risk of bias if more than one item was not fulfilled.

## Statistical analysis

We performed a random-effects dose-response meta-analysis to assess the relationship between changes in the duration of music and the changes in neonatal outcomes using the two-stage mixed-effect meta-analysis model. This method uses the "drmeta" command in Stata software version 17 (Stata Corp LLC, Texas, US). Greenland and Orsini described this method in 2006 [16]. This model used linear, quadratic, and cubic spline dose-response meta-analysis to evaluate the relationship between music and neonate status. We must note that we used restricted cubic splines of music therapy with three knots at fixed percentiles (25%, 50%, and 78%), having no a priori assumptions regarding the shape of the association. So, different models were fitted to the data, and the best model was chosen based on the Akaike information Criteria (AIC) and Bayesian Information Criteria (BIC).

We extracted the mean and standard deviation of neonatal outcomes in each arm of RCT that was measured based on the minutes. Then, we defined the mean difference in neonate outcomes after the intervention as the difference at the end minus the corresponding baseline value in the active and control arms of the trial. For this dose-response meta-analysis, the total duration of the music therapy per day was set as the midpoint in each category. We used the Egger test [17] and funnel plot [18] to examine the possibility of publication bias. The software Stata version 17, and Review Manager 5.4 were used for data analysis.

## Ethical considerations

This study has been approved by the Research Ethics Committee of Hamadan University of Medical Sciences with ethical ID: IR.UMSHA.REC.1403.589 and research ID: 140308227248.

## Results

### Description of studies

The PRISMA (Preferred Reporting Items for Systematic Review and Meta-Analysis) literature search flowchart is presented in Fig 1 and S1 Checklist. We retrieved 495 titles and abstracts, and after their screening, only 66 articles met our inclusion criteria. Finally, after we read their full text, only 30 randomized clinical trials remained in the final analysis. Reasons for the exclusion of the articles were trial protocol (n = 3), non-randomized trial (n = 2), wrong research sample (n = 2), non-physiological or behavioural outcomes (n = 8), effect size are reported as figures, not numerical value (n = 7), review, book, commentary, or editorial articles

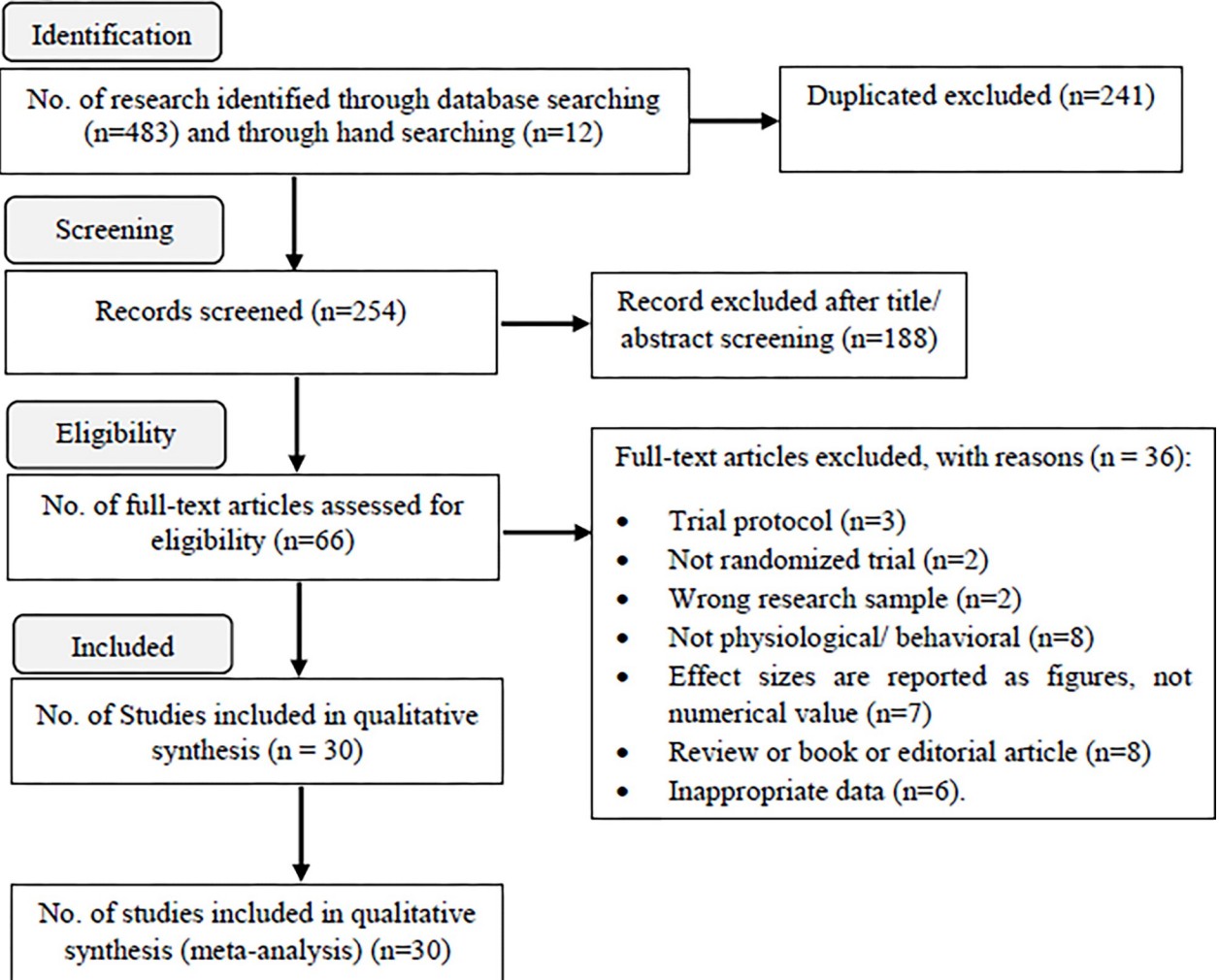

**Fig 1. Flow chart of systematic literature search for trials, published through Juan 28, 2024, that met the study inclusion and exclusion criteria.**

(n = 8), and inappropriate data (n = 6). Table 1 presents selected characteristics of the 35 trials included in the analysis. The list of articles that were excluded after a thorough review of their full texts, along with the reasons for their exclusion, can be found in S1 Table. We investigated 824 neonates in the intervention group and 1031 cases in the control group. Among the included trials, there were 22 studies in Asia, 2 studies in America, 5 studies in Europe and 1 studies in Australia. All trials reported adjusted effect sizes and were conducted in both sexes. The neonate's outcomes distribution was as follows: heart rate (24 trials), respiratory rate (15 trials), O2 saturation (25 trials), pain score (13 trials), body temperature (2 trials), behavioral score (6 trials), systolic and diastolic blood pressures (2 trial), and sleep duration (2 trial). The characteristics of the included studies are given in Table 1.

## Dose-response meta-analysis

With the increase of each minute of music therapy, the average of respiratory rate, pain score, SBP, DBP, behavioral score, and body temperature decrease by -0.353 (95% CI: -0.409 to -0.297), -0.153 (95% CI: -0.169 to -0.136), -0.307 (95% CI: -0.497 to -0.118), -0.089 (95% CI:

**Table 1. Description of eligible studies reporting the effect of music therapy on neonatal outcomes.**

| First author, year* | country | Sample size (I/C)§ | Outcomes | Type of music | Gestational age | Intervention characteristics |
|---|---|---|---|---|---|---|
| Alemdar, 2017 [32] | Turkey | 34/68 | Pain score, SaO2¶ | Intrauterine sounds | premature infants with gestational age 28–36 weeks | 1 day, one time in each day, 30 minutes in each time. |
| Alipour, 2013 [29] | Iran | 30/60 | Heart rate, Respiratory rate, SaO2, Behavioral Score | Lullaby music | premature infants with gestational age 28–37 weeks | 2 days, one time each day, 10 minutes each time. |
| Arnon, 2006 [33] | Israel | 31/31 | Heart rate, Respiratory rate, SaO2, Behavioral Score¥ | Live music | premature infants with gestational age 32–38 weeks | 2 days, one time in each day, 30 minutes in each time. |
| Barandouzi, 2020 [34] | Iran | 33/87 | Pain score | Braham's lullaby music | premature infants with gestational age 32–35 weeks | 3 days, one time in each day, 30 minutes in each time. |
| Caparros, 2018 [35] | Spain | 11/11 | Heart rate, Systolic and diastolic blood pressure, Respiratory rate, SaO2 | Relaxing music | premature infants with gestational age 32–36 weeks | 8 days, one time each day, 20 minutes each time. |
| Dehghani, 2015 [36] | Iran | 27/26 | Heart rate, Respiratory rate, SaO2, Body temperature | Music therapy is only mentioned, no details are given. | premature infants with gestational age 32–38 weeks | 5 days, one time each day, 60 minutes each time. |
| Döra, 2021 [37] | Turkey | 22/44 | Heart rate, Respiratory rate, SaO2, Pain score | White noise and lullabies | premature infants with gestational age 32–36 weeks | 2 days, one time each day, 2 minutes each time. |
| R. Keith, 2009 [38] | USA | 22/22 | Heart rate, Respiratory rate, SaO2 | Schwartz music | premature infants with gestational age 32–38 weeks | 1 day, one time each day, 18 minutes each time. |
| Kurdahi Badr, 2016 [39] | Lebanon | 42/84 | Heart rate, Respiratory rate, SaO2, Pain Score | Lullaby music or the music mothers listen to in the last trimester of pregnancy | premature infants with gestational age 28–36 weeks | 3 days, one time in each day, 10 minutes in each time. |
| Namjoo, 2021 [40] | Iran | 30/60 | Heart rate, SaO2, Sleep duration | Mother's live lullaby and recorded lullaby | premature infants with gestational age 28–36 weeks | 5 days, one time each day, 20 minutes each time. |
| Qolizadeh, 2018 [41] | Iran | 32/32 | Heart rate, Respiratory rate, SaO2, Body temperature | Holy Quran | Unclear | 2 days, one time in each day, 30 minutes in each time. |
| Ranger, 2018 [30] | Germany | 10/10 | Heart rate, Respiratory rate, Pain score | Live music (The music intervention was performed by a psychologist who had studied hand harp music play in the NICU.) | premature infants with gestational age 28–36 weeks | 2 days, one time in each day, 15 minutes in each time. |
| Uematsu, 2018 [42] | Japan | 15/13 | Heart rate, SaO2, Pain score | Braham's lullaby music | premature infants with gestational age 28–35 weeks | 10 days, one time in each day, 30 minutes in each time. |
| Tandoi, 2014 [43] | Italy | 17/17 | Heart rate, Respiratory rate, SaO2, Pain score, Behavioral Score δ | The original sound includes an original track composed of different sounds such as fetal heartbeat, breathing, blood flow, and ambience sounds | premature infants with gestational age 32–37 weeks | 3 days, three times in each day, 20 minutes in each time. |
| Amini, 2013 [44] | Iran | 25/25 | Heart rate, Respiratory rate, SaO2 | Lullaby or the music mothers listen and classical music including Mozart | premature infants with gestational age 28–36 weeks | 6 days, one time in each day, 20 minutes in each time. |

(*Continued*)

**Table 1.** (Continued)

| First author, year* | country | Sample size (I/C)§ | Outcomes | Type of music | Gestational age | Intervention characteristics |
|---|---|---|---|---|---|---|
| Arnon, 2014 [45] | Israel | 43/43 | Heart rate, Respiratory rate, SaO2, Behavioral Score | Lullaby or maternal singing | premature infants with gestational age 32–36 weeks | 2 days, one time in each day, 20 minutes in each time. |
| Dur, 2022 [46] | Turkey | 30/30 | Heart rate, SaO2, Pain Score | Classical music including Mozart | premature infants with gestational age 28–36 weeks | 1 day, one time each day, 1 minute each time. |
| Garunksteine, 2014 [47] | Lithuania | 35/35 | Heart rate, SaO2, Behavioral Score | Live & recorded lullabies | premature infants with gestational age 28–32 weeks | 1 day, one time each day, 20 minute each time. |
| Jabraieili, 2016 [48] | Iran | 25/25 | SaO2 | Brahms lullaby | premature infants with gestational age 29–34 weeks | 3 sessions, 15 minute each session. |
| Olischar, 2011 [49] | Australia | 10/10 | Sleep duration | Brahms lullaby | premature infants with gestational age ≥ 32 weeks | 1 day, one time each day, 20 minute each time. |
| Schlez, 2011 [50] | Israel | 52/52 | Heart rate, Respiratory rate, SaO2, Behavioral Score | Live harp music | premature infants with gestational age 32–37 weeks | 1 day, one time each day, 30 minute each time. |
| Taheri, 2016 [51] | Iran | 26/26 | Heart rate, SaO2, | Recorded male lullaby | premature infants with gestational age 29–38 weeks | 3 days, one time in each day, 40 minutes in each time. |
| Tang, 2018 [52] | China | 30/30 | Heart rate, SaO2, pain score | Lullabies & nursery rhymes, and more than 10 pieces of children's music with slower tempo that selected from "Chinese children music library" | premature infants with gestational age 28–36 weeks | 1 day, one time each day, from 10 minutes before peripherally inserted central catheter puncture and continued until 10 minutes after operation completion. |
| Wipple, 2008 [53] | USA | 14/13 | Heart rate, Respiratory rate, SaO2 | Lullaby music | premature infants with gestational age 32–37 weeks | Unclear but reported for before and during procedure |
| Lai, 2006 [54] | Taiwan | 15/15 | Heart rate, Respiratory rate, SaO2 | Instrumental lullaby and aboriginal Taiwanese lullaby | premature infants with gestational age < 37 weeks | 3 days, one time in each day, 60 minutes in each time. |
| Midilli, 2022 [55] | Turkey | 20/19 | Pain score | Braham's lullaby music | premature infants with gestational age 35–41 weeks | 3 days, one time in each day, 1 minutes in each time. |
| Mirzaei, 2022 [56] | Iran | 18/18 | Heart rate, SaO2 | Piano lullaby and Gary Stadler | premature infants with gestational age < 36 weeks | 1 day, one time each day, 5 minute each time. |
| Shah, 2017 [57] | Australia | 35/35 | Heart rate, SaO2, pain score | The "Deep Sleep" track from "Bedtime Mozart: Classical Lullabies for Babies" | premature infants with gestational age > 32 weeks | 1 day, one time each day, 27 minute each time. |
| Shukla, 2018 [58] | India | 50/50 | Pain score | Instrumental Indian classical flute music | premature infants with gestational age 28–36 weeks | 1 day, one time each day, 10 minute each time. |
| Yarahmadi, 2024 [59] | Iran | 40/40 | Heart rate, SaO2, pain score | Intrauterine sounds | premature infants with gestational age 28–36 weeks | 1 day, one time each day, 12 minute each time. |

*: All trials were conducted in both sexes.

ʃ: Oxygen saturation.

§: I (sample size in intervention group/music therapy arm); C (sample size in control group).

¥: Behavioral states graded on the behavioural state instrument (BSI). Six different behavioural states were distinguished: State 1 quiet sleep, State 2 active sleep, State 3 drowsy, State 4 quiet awake, State 5 active awake and State 6 crying.

δ: The behaviour score in this trial meant the "calming/relaxing" effects of music (more details not available in the method of the article).

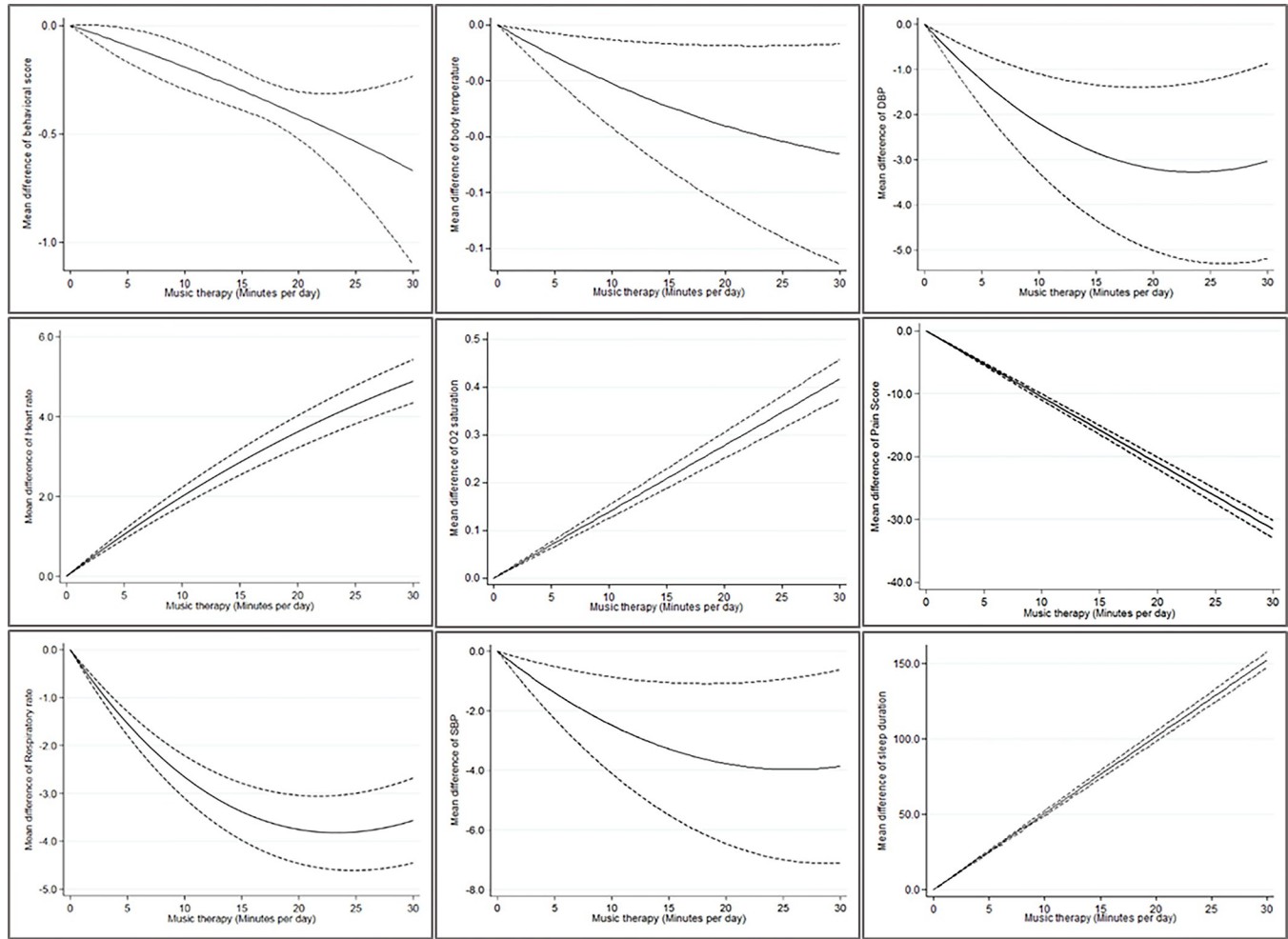

**Fig 2. Dose-response meta-analysis between music therapy and neonatal outcomes in a random-effects dose-response model.** The solid line indicates the mean difference, and long dashed lines indicate its 95% confidence interval.

-0.287 to 0.108), -0.027 (95% CI: -0.031 to -0.012), and -0.002 (95% CI: -0.004 to -0.0007). On the other hand, for each minute increase in music therapy, the mean difference (mean in the group receiving music minus the group not receiving it) of respiratory rate, pain score, SBP, DBP, behavioral score, and body temperature decrease by 35.3 beats per minutes, 15.3 VAS, 30.7 mmHg, 8.9 mmHg, 2.7, and 0.27°C. Also, with the increase of every minute of music therapy, the mean difference of O2 saturation, heart rate and sleep duration increase by 0.017 (95% CI: 0.013 to 0.021),0.892 (95% CI: 0.868 to 0.917), and 5.081 (95% CI: 4.910 to 5.253). On the other hand, with each minute increase in listening to the music, the mean difference (mean in the group receiving music minus the group not receiving it) of O2 saturation, heart rate and sleep duration increase 1.7%, 89.2 beats per minutes and 5.081 minutes per day, respectively (Fig 2).

## Methodological quality and risk of bias for included trials

As shown in the risk of bias assessment tool in Fig 3, the methodological quality of the included studies was acceptable. All studies included in this meta-analysis were classified as low-risk categories regarding incomplete outcome data, selective reporting, and other biases.

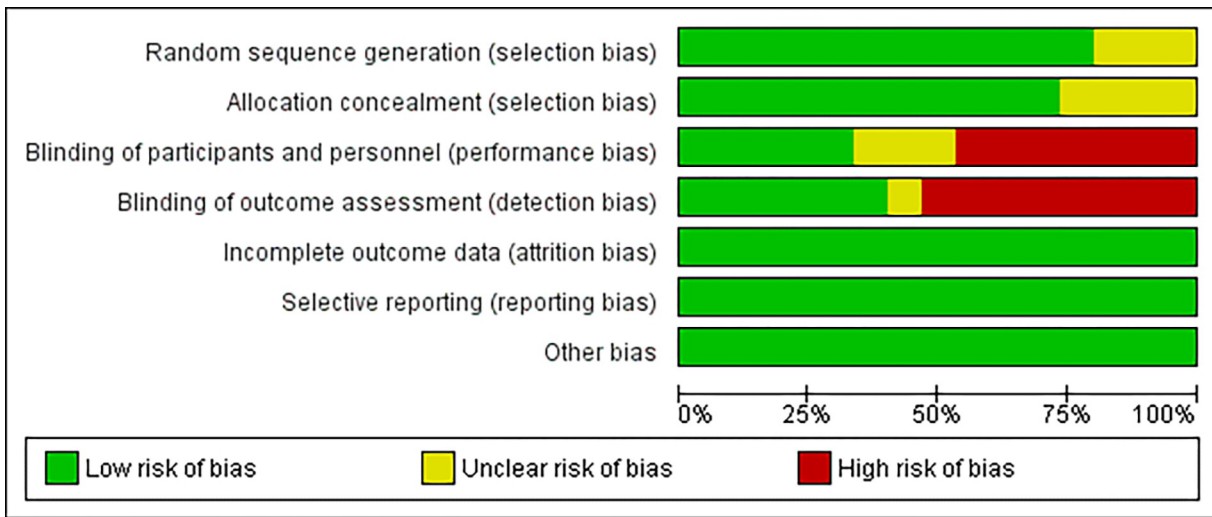

**Fig 3. Review the author's judgment about each risk of bias item as percentages across all included studies using Cochrane Collaboration's Risk of Bias Tool.**

As for the random sequence generation of included studies, about three-quarters of included trials mentioned generating a randomized sequence. Allocation concealment was not mentioned in twelve studies, but the rest of the studies used methods for generate the random sequence in which concealment was well done. Because of the nature of music therapy intervention, less than half of the trials successfully conducted the double-blinding methods. The completed risk of bias and quality assessments for each study are presented in S1 Fig.

## Publication bias

In our study, the Egger test result indicated no significant publication bias in included trials that investigated the effect of music therapy on neonatal outcomes (p = 0.294). There was no significant publication bias when we measured the Egger test for measured results. The significance values of the Egger test for included outcomes were as follows: behavioral score: 0.052, DBP: 0.317, heart rate: 0.083, O2 saturation: 0.925, pain score: 0.988, respiratory rate: 0.227, SBP: 0.317, sleep duration: 0.207, and body temperature: 0.534. Also, we assessed publication bias using the funnel plot. The symmetry distribution of studies around the null line in all funnel plots indicates the absence of publication bias (Fig 4).

## Discussion

This dose-response meta-analysis demonstrated that increased music therapy per minute is significantly associated with improved O2 saturation, heart rate, respiratory rate, sleep duration, body temperature, and systolic and diastolic pressure in preterm babies. This evidence shows a dose-response relationship between music therapy and neonatal outcomes. When a dose-response relationship is present, it adds plausibility to a causal relationship between exposure and outcome [19]. The key findings in both physiological and non-physiological areas will be the focus of our discussion, as detailed below.

### Physiological parameters

This meta-analysis highlights the clinical value of physiological parameters—key measurable characteristics of living organisms that describe the functions and processes occurring within

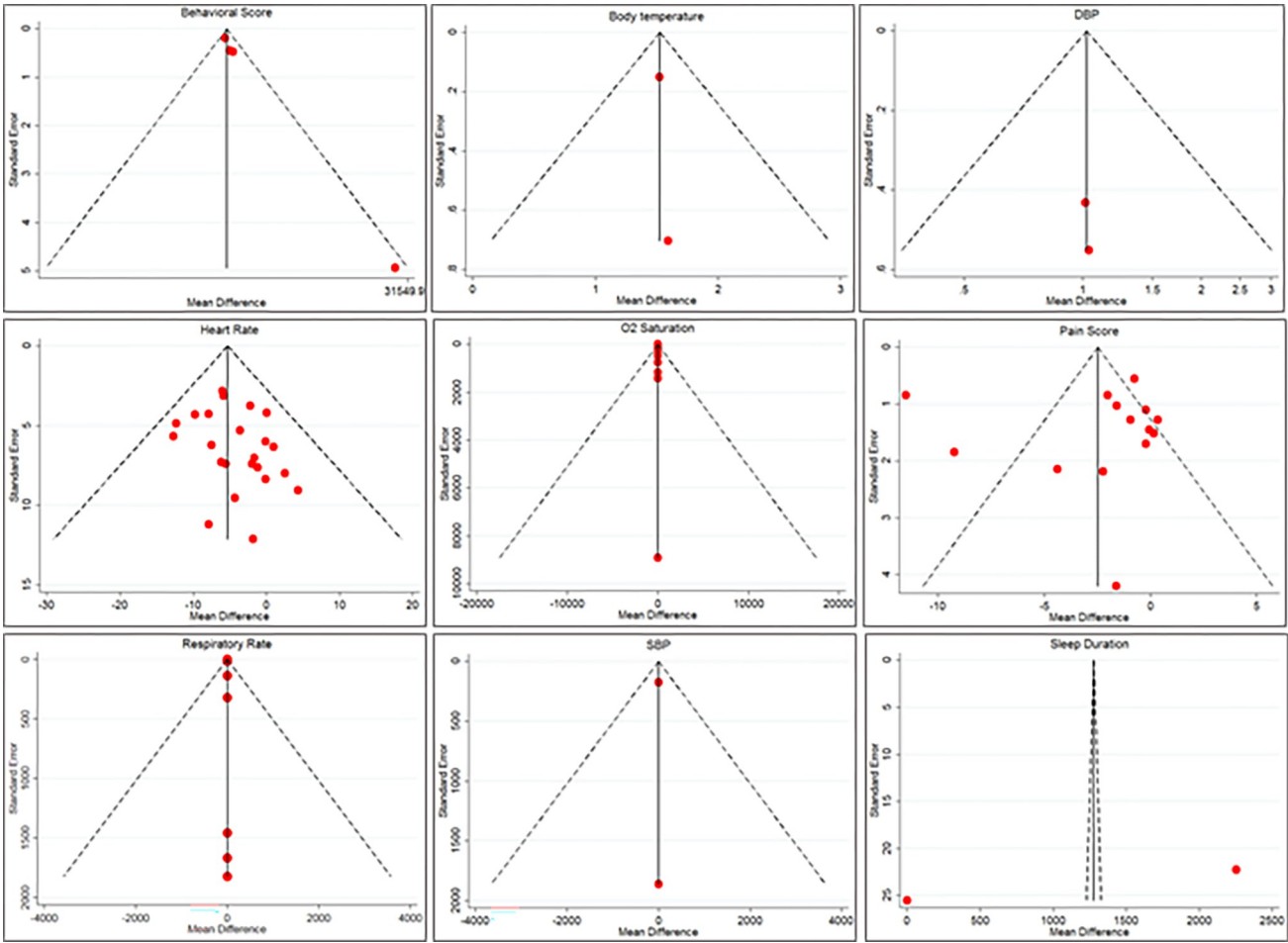

**Fig 4. Funnel plots of included studies by neonatal outcomes.**

the body. These parameters included O2 saturation, heart rate, respiratory rate, blood pressure, and body temperature, particularly in preterm infants. The following offers a justification for these findings. Research in music and neuroscience indicates that music fosters neurobiological processes and influences synaptic plasticity, neuronal learning, and brain readjustment [20]. This effect includes the activation of limbic and paralimbic structures in the human brain [21]. Particularly in newborns, music stimulates neuronal activation, which may contribute to improved physiological health.

With premature birth, the ideal intrauterine environment designed for optimal fetal growth and maturation is abandoned prematurely. Besides other stressful experiences, such as the separation from the mother and painful procedures, preterm infants must cope with the unusual sound environment of an intensive care unit [20]. Proper NICU levels are often 50–80 dB, sometimes reaching a peak of 120 dB [22]. However, the American Academy of Pediatrics recommended that sound levels not exceed 45 dB for neonatal intensive care units [23]. Preterm infants are susceptible to noise because their auditory system and brain development are in a critical, vulnerable, and fast period of growth. Stressful noise can further initiate stress responses in preterm infants. The sympathetic autonomic nervous system and the hypothalamic-pituitary-adrenal axis of the endocrine system are activated, known as the "fight or flight" reaction. This response may use energy reserves crucial for preterm infants' brain

development [24]. Thus, the overwhelming auditory neonatal intensive care environment is assumed to interfere with the short- and long-term neurobehavioral growth in preterm infants [25]. To summarize, the impact of music on physiological responses can be attributed to its influence on the nervous system, particularly the limbic and autonomic systems. This interaction triggers a relaxation response, which leads to regulated breathing patterns, a steadied heart rate, and increased oxygen saturation.

## Non-physiological parameters

The majority of published studies examining the effects of music in neonatal care focused on non-physiologic outcome parameters, including pain, behavioral state, and sleep pattern as evaluated by the infant's parents.

The beneficial impact of music therapy on enhancing the sleep duration of NICU infants can be understood as follows. Infants born preterm experience lighter and more active sleep compared to those born at term. Sleep patterns can be particularly important for preterm infants, who frequently show signs of bio-behavioral disorganization during daytime activities [26]. A study conducted by Schwichtenberg and colleagues found that preterm infants seemed to benefit from taking more "breaks" (i.e., naps) throughout the day, which helped them reorganize and sustain their engagement with the social environment [27]. Since music therapy induces alpha brain waves during wakefulness, it enhances sleep duration in preterm infants by positively impacting brain activity [26]. In this context, music therapy in neonatal care adopts a family-integrated approach, encouraging each family to discover their unique way of connecting with their newborn through music. In that case, it may empower parents by improving their well-being, boosting their self-confidence, and enriching the quality of their interactions with their newborn [20].

In our meta-analysis, we found that music therapy had a statistically significant effect on reducing pain and improving the behavioral state of premature infants. According to research, premature infants are even more sensitive to pain than older infants. Experiencing frequent painful stimuli at the start of their lives can lead to long-term consequences, including behavioral changes and potential susceptibility to psychosomatic problems and mental disorders in the future [11]. Music intervention is one of the many types of care that can change the environment to enhance health and well-being [28]. The significant impact of music on reducing pain and behavioral scores may be attributed to the fact that the studies examining the effects of music therapy were conducted during invasive procedures, such as blood sampling. This approach has been shown to effectively mitigate these two variables in preterm infants by improving physiological responses [29, 30].

## Future directions

This study reported a positive effect of music in neonatal care on physiological and non-physiological outcomes. However, concerns regarding statistical power and generalizability were raised by the majority of studies that reported positive effects, which had a sample size of 30 or less subjects. Another considering finding of this synthesis was that no trial had been conducted on the effectiveness of music therapy in improving the physiological and behavioral conditions of preterm infants in the African region. This is while according to the latest global burden of diseases study in 2019, the highest incidence of neonatal preterm birth was in sub-Saharan Africa [31]. Therefore, it is recommended that randomized clinical trials investigate the effectiveness of music therapy on neonatal outcomes in the African region.

## Limitations

This meta-analysis had several limitations and potential flaws. First, some studies seemed potentially eligible for inclusion in our meta-analysis, but we needed help accessing their full text. This problem can increase the possibility of selection bias. Second, in the randomized clinical trials included in our meta-analysis, some participants discontinued the study due to the length of follow-up. This can lead to a selection bias in our results. Third, the RCTs' varying music therapy modalities (from live music to recorded music) may affect the pooled results. The most important strength of this study is that it allows us to know how much listening to music improved neonatal outcomes in neonates. In addition, the broad search strategy used in this study increased the sensitivity of the search to include as many relevant articles as possible. Finally, an important point that should be mentioned is that our study examined more neonatal outcomes than Wei Yue's study in 2020 [10]. On the other hand, Wei's study was a simple meta-analysis, but ours was a dose-response meta-analysis.

## Conclusion

In conclusion, music therapy improves neonatal outcomes of O2 saturation, heart rate, respiratory rate, sleep duration, body temperature, and systolic and diastolic pressure. The existence of a dose-response relationship between music therapy and the mentioned neonate outcomes can strengthen the scientific background for therapeutic intervention in the newborn intensive care unit (NICU) for the treatment of adverse neonatal outcomes. There is no association between music therapies and pain score and behavioral score.

## Supporting information

**S1 Checklist. PRISMA_2020_checklist.**
(DOCX)

**S1 Dataset. Minimal dataset.**
(RAR)

**S1 Fig. The detailed risk of bias assessment.**
(PNG)

**S1 Table. Characteristics of the excluded studies.**
(DOCX)

## Author Contributions

**Conceptualization:** Fatemeh Shahbazi, Minoo Dabiri-Golchin, Marjan Shahbazi.

**Data curation:** Fatemeh Shahbazi, Marzieh Fattahi-Darghlou, Marjan Shahbazi.

**Formal analysis:** Fatemeh Shahbazi.

**Methodology:** Fatemeh Shahbazi, Marzieh Fattahi-Darghlou, Samad Moslehi.

**Project administration:** Marjan Shahbazi.

**Software:** Fatemeh Shahbazi, Samad Moslehi.

**Validation:** Fatemeh Shahbazi, Samad Moslehi.

**Writing – original draft:** Fatemeh Shahbazi, Marzieh Fattahi-Darghlou, Samad Moslehi, Minoo Dabiri-Golchin, Marjan Shahbazi.

**Writing – review & editing:** Fatemeh Shahbazi, Minoo Dabiri-Golchin, Marjan Shahbazi.

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
