## [Decision Letter · Decision Letter 0]

12 Jun 2024

PONE-D-23-42926Effect of music therapy on adverse neonatal outcomes: A systematic review and Dose-Response Meta-analysisPLOS ONE

Dear Dr. Shahbazi,

Thank you for submitting your manuscript to PLOS ONE. After careful consideration, we feel that it has merit but does not fully meet PLOS ONE’s publication criteria as it currently stands. Therefore, we invite you to submit a revised version of the manuscript that addresses the points raised during the review process.

**ACADEMIC EDITOR: **

Reviewer#1 has substantial considerations. Please, read the comments carefully and revise the manuscript accordingly. 

We look forward to receiving your revised manuscript.

Kind regards,

Mona Dür, PhD, MSc

Academic Editor

PLOS ONE

- https://doi.org/10.1155/2022/9161074

- https://doi.org/10.1002/14651858.CD011878.pub2

In your revision ensure you cite all your sources (including your own works), and quote or rephrase any duplicated text outside the methods section. Further consideration is dependent on these concerns being addressed.

3. In the online submission form, you indicated that [All data are fully available If requested by the editor.]. 

Additional Editor Comments (if provided):

Reviewers' comments:

Reviewer's Responses to Questions

**Comments to the Author**

1. Is the manuscript technically sound, and do the data support the conclusions?

Reviewer #1: Partly

Reviewer #2: Yes

2. Has the statistical analysis been performed appropriately and rigorously? 

Reviewer #1: Yes

Reviewer #2: Yes

3. Have the authors made all data underlying the findings in their manuscript fully available?

Reviewer #1: No

Reviewer #2: Yes

4. Is the manuscript presented in an intelligible fashion and written in standard English?

Reviewer #1: Yes

Reviewer #2: Yes

5. Review Comments to the Author

Reviewer #1: Reviewer comments

1. The results in abstract needs to convey more clarity.

2. “If the full text of the paper was unavailable, we received it from Sci-Hub, free paper, Research Gate, or Send an email to the corresponding author”. This line should be removed.

3. Kindly clearly clarify the primary and secondary outcomes.

4. Kindly revisit the title since it does not convey exact meaning i.e. physiological parameters are not adverse outcomes.

5. Kindly attach the filled PRISMA checklist.

6. Kindly use the PRISMA flow diagram template and redo the study flow diagram.

7. Publication bias: funnel plot should be generated and attached.

8. search strategy use may miss few of the RCTs since it advisable to screen the article manually rather than relying on the search strategy.

9. Explain the need for this meta-analysis in introduction section.

10. The search strategy gave me 306 results on pubmed itself. So I believe there are more article that can be found and made useful and increase the comprehensiveness of the article.

11. I think the statistics can be made useful if time is translated to days or hours. Giving the fractional numbers may not appeal as applicable.

12. How were the confounders accounted to highlight true results of the music therapy?

Response to criteria:

1. The study presents the results of original research.

Response: Yes

2. Results reported have not been published elsewhere.

Response: Not sure

3. Experiments, statistics, and other analyses are performed to a high technical standard and are described in sufficient detail.

Response: Needs lot of improvement.

4. Conclusions are presented in an appropriate fashion and are supported by the data.

Response: It needs to be reframed to convey correct meaning. The results done mean improvement but change in the physiological parameters.

5. The article is presented in an intelligible fashion and is written in standard English.

Response: Yes

6. The research meets all applicable standards for the ethics of experimentation and research integrity.

Response: Yes

7. The article adheres to appropriate reporting guidelines and community standards for data availability.

Response: Not sure. They should attach PRISMA checklist.

Reviewer #2: The authors made a great effort by using dose response meta analysis by comparing to establish the link between music therapy and neonatal outcomes of O2 saturation, heart rate, respiratory rate, sleep duration, body temperature, and systolic and diastolic pressure.

This dose-response meta-analysis has provided a reinforced cause and effect relationship between music and neonatal outcomes based on the observation of studies conducted in the past years.

6. PLOS authors have the option to publish the peer review history of their article (what does this mean?). If published, this will include your full peer review and any attached files.

Reviewer #1: **Yes: **Anurag Fursule

Reviewer #2: No

---

## [Author Response · Author response to Decision Letter 0]

27 Jul 2024

Reviewer #1: Reviewer comments

1. Comment: The results in abstract needs to convey more clarity. Response: We rewrite result in abstract.

2. Comment: “If the full text of the paper was unavailable, we received it from Sci-Hub, free paper, Research Gate, or Send an email to the corresponding author”. This line should be removed. Response: We have deleted this phrase according to your opinion.

3. Comment: Kindly clearly clarify the primary and secondary outcomes. Response: Dear reviewer, we have re-written the first paragraph of the method in more detail according to PICOS criteria. In the outcome subsection, we mentioned the primary and secondary outcomes.

4. Comment: Kindly revisit the title since it does not convey exact meaning i.e. physiological parameters are not adverse outcomes. Response: Many thanks for your thoughtfulness. We changed the title to " Effect of music therapy on behavioral and physiological neonatal outcomes: A systematic review and Dose-Response Meta-analysis".

5. Comment: Kindly attach the filled PRISMA checklist. Response: Dear Professor Anurag Fursule, we have attached the completed PRISMA checklist.

6. Comment: Kindly use the PRISMA flow diagram template and redo the study flow diagram. Response: Many thanks for your attention. We added PRISMA flow diagram template in figure 1. 

7. Comment: Publication bias: funnel plot should be generated and attached. Response: A funnel plot by neonatal outcomes was added to the result (Figure 3).

8. Comment: Search strategy use may miss few of the RCTs since it advisable to screen the article manually rather than relying on the search strategy. Response: We updated our search strategy and also used manual search. With this approach, 44 articles were finally included in our meta-analysis.

9. Comment: Explain the need for this meta-analysis in introduction section. Response: We are very grateful for your comment because it made us completely rewrite the introduction and write it more carefully and powerfully.

10. Comment: The search strategy gave me 306 results on PubMed itself. So I believe there are more article that can be found and made useful and increase the comprehensiveness of the article. Response: You enriched our article with your valuable comment. We updated our search strategy and also used manual search. With this approach, 44 articles were finally included in our meta-analysis.

11. Comment: I think the statistics can be made useful if time is translated to days or hours. Giving the fractional numbers may not appeal as applicable. Response: Dear reviewer, considering that in most studies, the entire period of inversion is less than 1 hour, minutes have a better meaning than hours. Using days instead of minutes is not very interesting. Because there may be two studies in which the length of the intervention period is 3 days in both of them. But the first one should play music for 20 minutes a day and the second one should play music for 5 minutes a day.

12. Comment: How were the confounders accounted to highlight true results of the music therapy? Response: The articles included in this meta-analysis were randomized clinical trials. According to epidemiology textbooks (such as Gordis epidemiology, chapter 10, pages 197-215 & Epidemiology beyond the basics, chapter 5, pages 175-202), randomization in clinical trial studies makes the compared groups balanced in terms of known and unknown confounders. In other words, randomization is a method of controlling confounding variables in clinical trials. Thank you for your valuable comment. Since I thought that the same concern might arise for the readers, I added a paragraph with the subtitle of confounding control in the method and explained this issue.

Reviewer #2: Comment: The authors made a great effort by using dose response meta-analysis by comparing to establish the link between music therapy and neonatal outcomes of O2 saturation, heart rate, respiratory rate, sleep duration, body temperature, and systolic and diastolic pressure. This dose-response meta-analysis has provided a reinforced cause and effect relationship between music and neonatal outcomes based on the observation of studies conducted in the past years. Response: Dear reviewer, thank you for your attention to our article.

---

## [Editor Report · Decision Letter 1]

14 Aug 2024

PONE-D-23-42926R1Effect of music therapy on behavioral and physiological neonatal outcomes: A systematic review and Dose-Response Meta-analysisPLOS ONE

Dear Dr. Shahbazi,

Thank you for submitting your manuscript to PLOS ONE. After careful consideration, we feel that it has merit but does not fully meet PLOS ONE’s publication criteria as it currently stands. Therefore, we invite you to submit a revised version of the manuscript that addresses the points raised during the review process.

**ACADEMIC EDITOR: **The revision of the manuscript has improved its overall quality. However, Reviewer 1 has several comments which should be addressed in the revised version. 

We look forward to receiving your revised manuscript.

Kind regards,

Mona Dür, PhD, MSc

Academic Editor

PLOS ONE

Additional Editor Comments:

Reviewer 2 did not send any comments. However, Reviewer 1 has several comments which should be considered and addressed in a revised version.

---

## [Author Response · Author response to Decision Letter 1]

17 Aug 2024

Dear editor, 

we have corrected all the comments of reviewer number 1 and sent back to you the revised article in the same format as you said.

---

## [Decision Letter · Decision Letter 2]

21 Oct 2024

PONE-D-23-42926R2Effect of music therapy on behavioral and physiological neonatal outcomes: A systematic review and Dose-Response Meta-analysisPLOS ONE

Dear Dr. Shahbazi,

Thank you for submitting your manuscript to PLOS ONE. After careful consideration, we feel that it has merit but does not fully meet PLOS ONE’s publication criteria as it currently stands. Therefore, we invite you to submit a revised version of the manuscript that addresses the points raised during the review process.

We look forward to receiving your revised manuscript.

Kind regards,

Phakkharawat Sittiprapaporn, Ph.D.

Academic Editor

PLOS ONE

Journal Requirements:

Additional Editor Comments:

(1) The discussion part requires enhanced clarity. Please examine the composition and sequence of sentences, especially in that section.

(2) Please review Table 1, which contains the research studies included in your systematic review as well as the associated statistics. Table 1 lacks at least one of the articles cited in the reference list.

Reviewers' comments:

Reviewer's Responses to Questions

**Comments to the Author**

1. If the authors have adequately addressed your comments raised in a previous round of review and you feel that this manuscript is now acceptable for publication, you may indicate that here to bypass the “Comments to the Author” section, enter your conflict of interest statement in the “Confidential to Editor” section, and submit your "Accept" recommendation.

Reviewer #2: All comments have been addressed

Reviewer #3: All comments have been addressed

2. Is the manuscript technically sound, and do the data support the conclusions?

Reviewer #2: Yes

Reviewer #3: Yes

3. Has the statistical analysis been performed appropriately and rigorously? 

Reviewer #2: Yes

Reviewer #3: I Don't Know

4. Have the authors made all data underlying the findings in their manuscript fully available?

Reviewer #2: Yes

Reviewer #3: Yes

5. Is the manuscript presented in an intelligible fashion and written in standard English?

Reviewer #2: Yes

Reviewer #3: Yes

6. Review Comments to the Author

Reviewer #2: The meta-analysis done elaborately to demonstrate the music therapy on behavioral and physiological neonatal outcomes and the limitation. The revised manuscript has improved and acceptable for publication.

Reviewer #3: Dear Authors,

This systematic review brings highly relevant points related to music therapy and dose effect. Thank you for addressing them. It is something we need to look at in the near future for premature babies and other populations as well.

Additional comments are in a separate PDF file of your manuscript.

Overall the English narrative is clear, however it needs better clarity in the discussion section. Kindly review the writing / sentence order particularly in that section.

It will help if you kindly review Table 1 with the research studies included in your systematic review and potentially in the statistics. There is at least one article that is mentioned in the reference list, that it is not in Table 1. Please, see my comment on the PDF file.

Looking forward to your article.

7. PLOS authors have the option to publish the peer review history of their article (what does this mean?). If published, this will include your full peer review and any attached files.

Reviewer #2: **Yes: **Jagadeesh Ramasamy

Reviewer #3: No

---

## [Author Response · Author response to Decision Letter 2]

1 Dec 2024

Comment 1: The discussion part requires enhanced clarity. Please examine the composition and sequence of sentences, especially in that section. Reply: Dear referee, thank you for your valuable comment. We have rewritten the discussion and highlighted it in yellow.

Comment 2: Please review Table 1, which contains the research studies included in your systematic review as well as the associated statistics. Table 1 lacks at least one of the articles cited in the reference list. Reply: Thank you for your high accuracy. Rafael's article is the same as Caparros 's article, so we removed it from Table 1. Corresponding to this deletion, other parts of the article were corrected.

---

## [Editor Report · Decision Letter 3]

16 Dec 2024

Effect of music therapy on behavioral and physiological neonatal outcomes: A systematic review and Dose-Response Meta-analysis

PONE-D-23-42926R3

Dear Dr. Shahbazi,

We’re pleased to inform you that your manuscript has been judged scientifically suitable for publication and will be formally accepted for publication once it meets all outstanding technical requirements.

Kind regards,

Phakkharawat Sittiprapaporn, Ph.D.

Academic Editor

PLOS ONE

---

## [Editor Report · Acceptance letter]

27 Dec 2024

PONE-D-23-42926R3 

PLOS ONE

Dear Dr. Shahbazi, 

I'm pleased to inform you that your manuscript has been deemed suitable for publication in PLOS ONE. Congratulations! Your manuscript is now being handed over to our production team.

Kind regards, 

on behalf of

Dr. Phakkharawat Sittiprapaporn 

Academic Editor

PLOS ONE